# Barriers to Hepatitis B Screening and Prevention for African Immigrant Populations in the United States: A Qualitative Study

**DOI:** 10.3390/v12030305

**Published:** 2020-03-11

**Authors:** Catherine Freeland, Sierra Bodor, Udara Perera, Chari Cohen

**Affiliations:** 1Hepatitis B Foundation, 3805 Old Easton Rd., Doylestown, PA 18902, USA; Sierra.Bodor@hepb.org (S.B.); Chari.Cohen@hepb.org (C.C.); 2Drexel University Dornsife School of Public Health, 3215 Market Street, Philadelphia, PA 19104, USA; uwp23@drexel.edu

**Keywords:** hepatitis B, African immigrant, liver cancer, public health, qualitative research, barriers, religious value, cultural norms, stigma

## Abstract

Chronic hepatitis B infection (HBV) disproportionately affects African Immigrant (AI) communities in the U.S., with a reported infection rate of 15%. HBV screening rates within these communities are low. This study sought to better understand the socio-cultural determinants associated with low HBV screening among AI communities and identify potential strategies to help inform the development of effective HBV education and screening interventions. Seventeen in-depth interviews were conducted with community health experts working in AI communities throughout the U.S. Interviews explored the potential impact of culture, perception of health, awareness of HBV, religious practices, current screening practice, provider relationship, and behaviors towards general prevention. Interview data were analyzed using thematic analysis. Religious preferences and cultural norms affect health care access, perceptions towards prevention, awareness of HBV, and contribute to myths and stigma within this population. Participants reported a lack of HBV knowledge and awareness and barriers to health care access including, cost, language, racism, understanding of Western Medicine, and usage of traditional medicine. This study elucidates the role of religious and cultural beliefs as barriers to HBV screening and care. Results can contribute to public health efforts to increase awareness, screening and vaccination efforts within AI communities.

## 1. Introduction

Chronic hepatitis B virus (HBV) is a significant global health concern, with an estimated 292 million individuals affected worldwide [1]. According to the World Health Organization (WHO), between 20% and 30% of those chronically infected will develop life-threatening complications, including liver cirrhosis and hepatocellular carcinoma (HCC) [1,2,3]. The highest rates (>5%) of HBV are within Africa, and the Western Pacific regions [1], and an estimated 54,000 people with HBV immigrate to the United States (U.S.) annually [4]. While the actual burden of HBV in the U.S. is unknown due to lack of routine screening and disease surveillance, estimates suggest up to 2.2 million chronically infected individuals reside in the U.S. [5,6]. Asian Americans and Pacific Islanders (AAPIs) are estimated to make up 50% of this infected population, while those from Africa and the Caribbean may make up an additional 15% [4,5,6,7,8,9,10,11,12].

Limited research has focused on African Immigrant (AI) communities in the U.S., contributing to inadequate prevalence data, prioritization, and awareness of HBV as a health burden for AI. In Africa and the U.S., most research has occurred within small-scale community-based settings. An international meta-analysis detailed pooled prevalence rates for chronic HBV; the highest prevalence rates were in Sudan (18.6%), Liberia (16.5%), Guinea (16.3%), Eritrea (15.5%) and Zimbabwe (13.9%). Within the same study, foreign-born persons in the U.S. from Africa had the highest average HBV prevalence rates of 10.3% compared to those from Asia (7.3%) and Oceania (4.8%) [4]. Since 1980, the number of AIs in the U.S. has increased by over 750% [12].

HBV screening rates remain low among foreign-born communities in the U.S., including AI communities [10,13]. Limited knowledge and awareness of HBV among the general population is likely a contributing factor, as are multiple cultural, linguistic, and systemic barriers that are frequently associated with lower health care access in underserved communities [14]. These complex challenges are not well understood and prevent the development of effective strategies from improving HBV screening among AIs. In order to meet the WHO and National Academies of Science Engineering and Medicine (NASEM) goals of eliminating hepatitis B by 2030, high-risk AI community members must be appropriately screened for HBV, and those infected can access evidence-based care and prevention services such as vaccination and treatment [10,13]. The goal of this qualitative study was to better understand the socio-cultural determinants associated with low HBV screening among AI communities and identify potential strategies to help inform the development of more effective HBV education and screening interventions.

## 2. Materials and Methods

### 2.1. Participant Recruitment

Four investigators conducted 17 individual in-depth interviews with community health experts working in AI communities across the U.S. Through convenience sample, participants (*n* = 17) were recruited from the Coalition Against Hepatitis for People of African Origin (CHIPO), a national coalition working to address HBV disparities in AI communities, and included community health workers, researchers, and physicians. In order to meet the inclusion criteria, participants had to have experience conducting health research, provide health care services, programming, or community outreach for the AI community. Additionally, participants had to identify as being of African descent and be fluent in English. Once participants were identified as meeting the inclusion criteria, one-hour telephone interviews were recorded and completed at the participants’ convenience. All participants provided verbal consent to participation and were compensated with a gift card for their participation.

### 2.2. Data Collection and Qualitative Analysis

A literature review was conducted to identify constructs of importance for the interviews. From these findings, a semi-structured interview guide was designed and used to facilitate telephone interviews (*n* = 17) (Appendix A). A 13-question interview guide was developed based on a literature review and expert consultation. The interview guide was aimed at gaining information on general health care barriers and those specific to HBV awareness and screening. The interview was designed to understand the perspectives and barriers surrounding HBV within the AI community. Questions explored the potential impact of culture, perception of health, awareness of HBV, religious practices, providers and traditional medicine, and behaviors towards general prevention and hepatitis B screening behaviors. Questions also focused on the most trusted and accessible sources of health information for AI communities. Each 60 min interview was conducted by phone, audio-recorded, and transcribed. The interview data were analyzed using thematic analysis [15].

Thematic codes were developed in two ways: (1) by literature and interview guide, and (2) inductive reasoning of a subsample of interview transcripts. Four trained members of the research team independently evaluated the interviews for coding. Each interview was assigned a primary and secondary coder. Discrepancies in coding were resolved in consensus and review with members of the research team. Key informant quotes were extracted from the interview data that represented the codes as defined by the codebook.

### 2.3. Ethics

This project was reviewed by Fox Commercial IRB, Ltd., and approved for exemption, as the research involved the use of interview procedures for which study participants could not be identified. No identifiable information was collected or included in the study.

## 3. Results

### 3.1. Participants

A total of 17 participants completed interviews, —11 women and six men. All interviews were conducted in the English language. Those interviewed were from nine states across the U.S., representing community-based and non-profit organizations (*n* = 8), federal/state organizations and health departments (*n* = 4), academic and research institutions (*n* = 2), a federally qualified health center (FQHC, *n* = 1), physicians (*n* = 1) and individual community members (*n* = 1). Participants were located in major U.S. cities throughout the United States, where large African communities are located, including Augusta, ME, Boston, MA, Houston, TX, New York, NY, Philadelphia, PA, Rochester, MN, Seattle, WA, and Washington, D.C. Each individual had extensive experience in serving and providing services for AI communities from Sub-Saharan Africa. Specific populations served include Somalia, Ethiopia, Nigeria, Ghana, Sudan, Chad, and South Africa, among others. Saturation was reached after ten interviews, but seven additional interviews were collected to ensure the information was comprehensive.

The coding of the interview transcripts revealed two overarching themes impacting AIs’ barriers to health care in regard to HBV: (1) religious values and (2) cultural norms. Within the two overarching themes, it is essential to highlight the diversity of culture and religion across the African continent, which was identified and described within interviews. Each country and each community within countries represents diverse cultural and religious practices, which may not be generalizable for the entire African continent, but the overarching themes are significant contributors to how health is viewed. Subthemes identified were (1) health care access, (2) perception towards prevention, (3) HBV awareness, (4) HBV myths, and (5) stigma. Subthemes were interconnected with cultural norms and religious preferences. Both overarching themes and subthemes are described below and presented in Figure 1.

### 3.2. Overarching Theme: Religious Values

All advisors emphasized religion in the African context as deeply rooted in community values. Interviewees conveyed that religious values affect health care access and utilization within the AI community. When it comes to testing or screening for specific diseases like HBV, many interviewees stated that community members do not want to get tested due to a belief that everything, including health, is “God’s will.” One interviewee stated, *“People may not want to get tested because they believe that everything is God’s will, as well as thinking that it is better to not know.”* Other interviewees echoed that many Africans tend to see sickness as something that is inflicted by God in the form of punishment for wrongdoing or immorality. One participant stated; *“The person (infected) may go to a pastor who says that ‘it is because you have sinned, that is why you have been affected by this disease and therefore spiritual (healing) is the way to go.’”* This belief also contributed to the belief that people can be healed through spiritual means. Participants stated about community members: *“If you believe in God then you should allow him to heal you first when all else fails*–*that is when you go to the doctor,”* and that people often *“Pray about it (health issue) or go to their spiritual leader before going to medicine.”* Thus, community members might seek healing at a religious institution or spiritual healer before engaging with a health care provider when they are sick. It was repeated by many interviewees that religious preferences affect health care access, perceptions towards prevention, knowledge of HBV, and contribute to myths and stigma within this population.

### 3.3. Overarching Theme: Cultural Norms

Interviewees commonly mentioned culture within the AI community as being an essential factor influencing health care access and HBV testing. Many mentioned that in some African cultures, discussing disease is taboo, where health is something to *“Be kept quiet about.”* One interviewee mentioned specifically to HBV, *“… It is like a secret, and you don’t talk about those kinds of health issues. Because it’s looked at as ‘you did something to get this disease.’”* Another mentioned the impact this has within this community, *“We’re seeing a lot of cancer deaths, but nobody’s talking about them. Families don’t want people to know when people die of cancer, so it’s kept a secret.”* Interviewees mentioned that even within families, a culture of secrecy around health is common. One participant explained, *“I feel like, death, or going and taking care of their health is never, it’s not important—it’s like the last thing they’ll think about, there are only a few people you’ll come across that would say you know, ‘my health is very important to me.’ It’s very, very rare that you hear someone say that to you.”*

The use of traditional and herbal medicine is common among African cultures, and many AI in the U.S. call home for traditional remedies before seeing a Western practitioner. Interviewees mentioned that commonly when someone becomes sick, the first thing they may do is call “back home” (to where they were born) to seek guidance or herbal treatments from family or spiritual healers. One advisor stated, “*We were just, we were looking at mental illness. And one of the things if someone is depressed, they’ll get their family in Somalia to go to the Imam or the traditional healer in Somalia and do rituals and that, because you know, it’s their home ground…some kind of cleansing, a restorative process. It depends so much on culture and their religion. It could be from killing a goat. Leading a chicken. Saying a prayer, lighting candles. And those, or drinking saltwater*.”

### 3.4. Health Care Access

Interviewees also noted the complexity of the U.S. health care system as a contributing factor related to health care access and testing for chronic diseases like HBV. They described challenges surrounding the U.S. health care system that included language barriers and health literacy, provider trust, racism, cost, and difficulties navigating a foreign health care system. According to interviewees, AIs often feel that the health care system is complex and challenging to navigate. Many AIs *“Do not understand how it works, so this withdraws them from seeking health care or going to the doctor.”* Frequently, AIs require translation services when visiting a provider, and often services are not reliable. One interviewee stated, *“For some communities, English is not even a second language, and some do not speak English at all.”* This can create challenges within the patient–provider relationship. In many African languages, there is not a word for hepatitis, further complicating the ability to convey health information accurately. Language barriers can contribute to a feeling of misunderstanding and cultural disconnect and lead to *“Resistance to utilizing health care and gaining awareness.”* In an attempt to bridge the language barrier, many AIs rely on their children to interpret, translate, and navigate the health care system during medical visits, which can be significantly challenging for sensitive topics and those disease states carrying stigma such as HBV.

In addition to language complications, provider trust is an issue within AI communities. Several interviewees noted historically rooted fears of experimentation and confidentiality concerns, leading to a general mistrust of medical providers. One interviewee stated that AI community members have concerns such as, *“What are they going to do with my blood when they take it, are they going to test me for something without my knowledge, and is my blood going to be used for an experiment?”* There is also a fear that a provider will send their medical information to their employer, which could result in a loss of employment or deportation. Fear around accessing care is also tied to word-of-mouth experiences; one interviewee mentioned that in her home country, anyone who went to the hospital *“Never came back, they died.”* Many participants also highlighted that community members fear their personal information becoming public so that their neighbors would know their HBV status. This fear of being labeled or “*Branded with a diagnosis*” within the community promotes continued silence around sickness and promotes stigma around HBV.

### 3.5. Perceptions Towards Prevention

Interviewees universally stated that health prevention behaviors, such as disease screenings, are not commonly practiced within the African cultural context. This is often explained by differing health care systems in many African countries and contributing to low HBV screening rates among AI in the U.S. Health care is most often symptom-driven rather than preventative, and with a silent disease like chronic HBV, symptoms do not usually appear until the end stages of liver disease, when it is often too late for health intervention. An interviewee stated, *“Preventive medicine isn’t something that culturally is commonly accepted… seeking health care is more symptom-driven or problem-driven that drives individuals to access health care and doctors.* Multiple interviewees noted that most AIs, *“Do not know they have it (HBV).”* It was also mentioned that it can be scary to go to the doctor for fear of a diagnosis, even when you did not feel sick: “*You go to the doctor and you feel fine and then all of the sudden ‘I have hepatitis B’. I think that is like, ‘How can you tell me I have hep B when I feel fine?’”* The majority of interviewees stated that the only reason someone might get screened for HBV is if they have basic knowledge about HBV and understand that they may be at risk.

### 3.6. HBV Knowledge

All seventeen interviewees described a low level of HBV knowledge among AI community members compared to other health issues like diabetes, hypertension, and HIV. One interviewee provided this example: *“Almost everyone is aware of HIV, everyone knows what HIV and AIDS are, but when you talk about hepatitis B, you have to explain to people what it really is.”* Most interviewees suggested that community members and providers do not realize AIs are at risk for HBV and should be screened. One interviewee stated, *“It’s interesting how most West Africans do not even know about this disease (HBV).”* Several others stated that many AIs do not realize there is even a test for HBV, and that *“There is a lack of knowledge or incorrect knowledge around how it’s acquired, how it’s transmitted, and what it means to have an infection.”* When asked about health and prevention one interviewee stated, *“I feel like a lot of them (AIs) are uneducated when it comes to health… once they’re aware of their hepatitis B—because I know many of them even wouldn’t prefer to get tested in order to find out just because they feel like it’ll be a burden for them to live with it. It’ll be terrible and depressing, and they feel that if they feel healthy, therefore there’s no need to go and see a doctor.”* This quote demonstrates the general feeling of wanting to avoid dealing with illness, especially one that may not be very well understood. One interviewee mentioned, *“For the most part, most people have never heard of hepatitis B, and don’t think they are at risk for it because they don’t think that it is an issue for Africans.”* Additionally, interviewees stated that members of the AI community are not aware that there is an HBV vaccine available, or that they need three doses of the vaccine to be fully protected. One interviewee pointed out that, *“Most West African born immigrants in their home countries at the time of their birth did not have access to vaccination programs like we do here.”*

### 3.7. HBV Myths

Interviewees emphasized that there are many myths and misconceptions associated with the transmission and cause of HBV within the AI community. One advisor recalled, *“I have seen people think that if a bat flies over your head you can get hepatitis that way… I have heard a lot of people say things like that.”* Another common misconception is that one can get HBV casual contact such as from hugging, kissing, and touching someone or “*Eating off someone else’s plate*.” Although in Africa, HBV is most commonly contracted during birth or early childhood, interviewees echoed that more commonly HBV is thought to be associated with moral wrongdoing, sexual promiscuity or drug use. Another common belief is that HBV is linked to voodoo or witchcraft: *“Some people, believe in–like special powers or witchcraft... and so, some sicknesses are attributed to that. And for those reasons people may not get care right away until, like I said, it gets to the point where the prayer isn’t working or whatever terrible thing you were given isn’t working. That’s when they may go to seek medical care.”* This highlights the complexities of beliefs and experiences that play a role in low screening and care rates for HBV.

### 3.8. HBV Stigma

Stigma has been shown to be a significant barrier to HBV screening for many cultures within AI communities. One interviewee stated, *“So diabetes (hypertension) is one that folks openly talk to me about, or if we’re doing a risk assessment, they’ll say they have diabetes or hypertension. They won’t talk about those stigmatizing diseases, like HIV, mental health–even hepatitis B is stigmatized, cancers are stigmatized in some ways. Folks will talk to me and say ‘Oh, my cholesterol is high,’ or things like that…It’s easier to get them to check their glucose, check their blood pressure, check their BMI, than to do HIV testing or other STD testing, or even to talk about a mental health assessment.”* Additionally, a theme throughout the interviews described the fear of being labeled as having HBV. One advisor mentioned his personal experience, *“I’ve had patients who had HIV. And when I was working in a hospital and there was one designated floor for the HIV clinic. So, you know, and everybody goes to that floor, you know what they’re there for most Africans were very, very, very, reluctant to go to that floor for fear of being seen by another community member.”* There is also a concern that once diagnosed, one may become socially isolated from friends and family due to HBV being frequently associated with individuals engaging in drug use or sexually promiscuous behavior. Many interviewees also noted that a positive HBV diagnosis indicates that an individual did something immoral. One explained, *“If you contract hepatitis B, it may not necessarily be accurate to you being exposed to infected blood, but because of some spiritual misdeeds that you have become infected… or maybe it is attributable to something your family has done.”*

### 3.9. Study Limitations

As with most qualitative studies, this study included a small sample size, which limits the generalizability of the findings. While the interviewees represented diverse backgrounds and experiences, diversity of the African continent is likely not adequately represented within this study, and more research on the barriers faced within the AI community should be further explored. This study only captures the opinions of seventeen experts working in HBV public health efforts in the U.S., and some information could be missing or not fully explored. Interviews from patients’ perceptions would further inform our work, and future research should focus on this.

## 4. Discussion

It is essential to note the complex and interconnected role of culture and religion on AIs’ views of health and utilization of health care services. This complexity between culture and religion was seen throughout each interview and, when added to beliefs about Western medicine, lead to significant barriers to HBV-related behaviors, including uptake of screening and vaccination. An important public health prevention strategy to consider for this population is improving access to vaccination for HBV. According to the Centers for Disease Control and Prevention (CDC), in 2017, the reported HBV vaccination coverage (>3 doses) was 25.8% for adults >19 years, and when broken down by ethnicity in the black population, findings showed coverage among adults aged 19–49 was at 30.7% [16]. Broken down by foreign-born community vaccination coverage, this shows protection among only 28.4% of this community [17]. While this study focused on assessing barriers related to HBV testing, more emphasis should be placed on improving prevention strategies and improving awareness and knowledge related to HBV vaccination in the AI community. 

Throughout the analysis, it was challenging to simplify and categorize themes independently as many were interrelated with one another. Both religion and cultural norms are essential to understanding the complexity of barriers within the AI community. When designing public health programming and interventions, it is crucial to be aware of the cultural and religious preferences of communities. Our interviewees noted that *“There is a thin line between religion and culture for most of our clients, where culture influences religion and religion influences culture. That all impacts behavior.”* In order to be successful in preventing HBV and identifying current infections, we have to be cognizant of these themes and barriers within future outreach strategies. Additionally, it is important to consider barriers to health care access for this population, including, cost, language, racism, this community’s understanding of Western Medicine and usage of traditional medicine. The lack of HBV knowledge may potentially leave individuals susceptible to infection. Overall, findings suggest that more effort towards improving knowledge surrounding HBV should be prioritized within this population to address misconceptions and lack of knowledge and future public health messaging should incorporate cultural norms represented by each community.

Many interviewees noted the lack of HBV programs specifically focused on the AI population. As the AI population is growing in the U.S., and we know that HBV disproportionately impacts AIs, it is critical that we design and implement public health programming to reduce barriers to HBV screening and care. Programs should improve HBV awareness, dispel common myths and misconceptions, reduce the stigma associated with HBV, and attempt to address systemic challenges to health care access while involving the community itself. Interviews demonstrated the fear of losing one’s community connection and social isolation after a disease diagnosis. This shows that the community has a significant impact on behavior related to a stigmatized disease like HBV. This also demonstrates the power of the community connection itself and the leverage the community can have on health. Utilizing the community itself for awareness and messaging can have a significant impact on addressing health challenges like HBV. The information we learned from this study can be used in the design of such programs for HBV in the AI community. However, it is essential to encourage future research and funding to better understand specific challenges and needs of AI communities so that we can improve HBV testing, prevention through vaccination, and care among this highly impacted community.

## 5. Conclusions

The WHO and NASEM recognize the global burden of HBV and has called for the elimination of HBV and hepatitis C by 2030 [13]. With 60 million estimated HBV infections in Africa [13] and an estimated 2.2 million in the U.S. [5,6], public health efforts must be specific to address barriers within AI communities, which are disproportionately affected by HBV. This study identified the complex social and structural barriers affecting HBV knowledge, screening, vaccination, and successful linkage to care within the AI community. The results of this study contribute valuable information to the currently limited literature documenting the role of religious and cultural beliefs on health care access within this community and can serve as a starting point to inform culturally-relevant public health interventions aimed at increasing HBV knowledge and awareness, screening and vaccination efforts. More research is needed to better understand some of the barriers identified within the AI community, and more funding and support to address these barriers at the public health level for HBV awareness and testing is a significant need in this community.

## Figures and Tables

**Figure 1 viruses-12-00305-f001:**
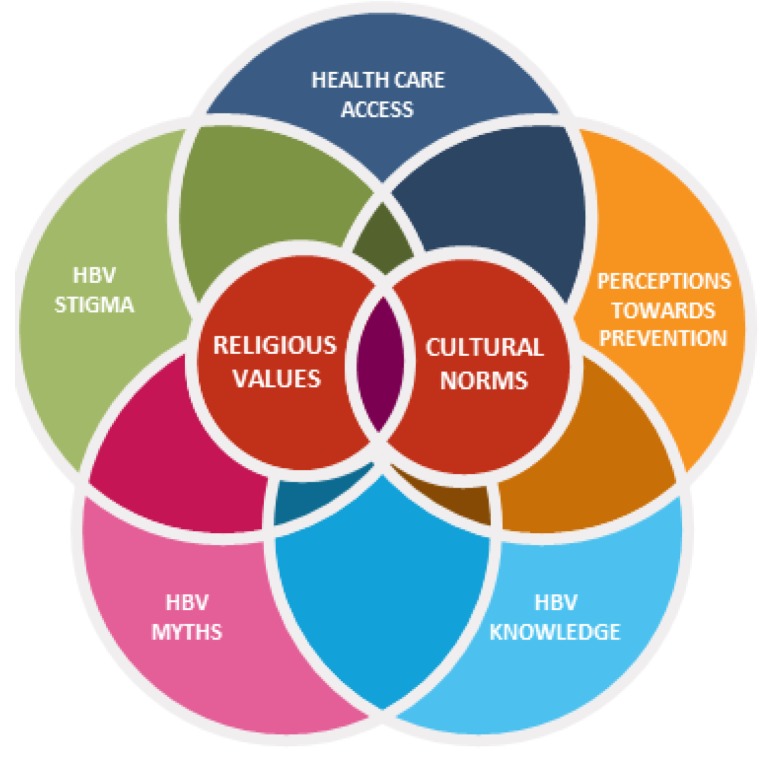
A visual depiction of the interconnectedness of the barriers faced by African Immigrant (AI) communities related to hepatitis B perceptions (red=the overarching themes, Religious Values and Cultural Norms; blue, orange, yellow, green, navy = are represented by the subthemes).

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
