# Peer review of "Barriers to Hepatitis B Screening and Prevention for African Immigrant Populations in the United States: A Qualitative Study"

_viruses, 2020, doi:10.3390/v12030305_

Round 1
Reviewer 1 Report
The paper by Freeland C. et al. aims to identify the social barriers to HBV screening and care in African Immigrants in the US. On my opinion, the paper is mostly of interest for social journals, more than for virology journals since it identifies the role of social and cultural beliefs in the lack of actions against hepatitis B infection by African immigrants in the US. Moreover, the data are based moslty on interviews by self-identified people as community helath workers and academic or research institution members, posing some doubts on the reliability of the obtained data.
Author Response
Response to reviewer 1: Thank you for your comment. While this reviewer did not specifically state adjustments to the article itself, we did add additional descriptions on the methods and recruitment sections to strengthen this paper. This reviewer noted that the data presented is based only on qualitative interviews, which is correct. Qualitative research has been used extensively in data collection and is used to better understand complex phenomena. Like quantitative research, qualitative research can address causation (Black, 1994). Unlike quantitative research, it seeks to understand the “what” question, not the “how often” (Black, 1994). A common methodology of collecting qualitative data can be through interviews, observations and focus groups among other methodology. In order to better understand what for this research we choose interviews for this data collection. The “what” within this paper is to understand barriers within certain populations to accessing care or for exploratory research to understand the impact of a viral infection like hepatitis B on a specific sub-population. This research allows public health professionals, physicians, and researchers to tailor public health programming, clinical interventions, and research design to address barriers identified within this study. Qualitative data is not generalizable data but allows for further understanding of a complex issue like the one presented in this paper. We hope that updates within the text of this paper in response to all reviewers provide more clarification and context.
Citation: Black, N. Why we need qualitative research. Journal of Epidemiology and Community Health 1994; 48: 425-426. https://www.ncbi.nlm.nih.gov/pmc/articles/PMC1060002/pdf/jepicomh00200-0001b.pdf
Reviewer 2 Report
Dear Freeland et al.,
In this paper, the authors conducted a qualitative study based on seventeen in-depth interviews with experts working for the African Immigrant communities and suggested major and minor barriers that restrict HBV screening and prevention for immigrant populations. Crossing the barriers will figure out the authentic number of immigrants infected with chronic HepB and offer a guidance when and how to treat existing HBV immigrates, which contributes to the aim, the elimination of Hep B by 2030 called by the WHO. In general, the paper is well organized and written, however several points require clarification before an acceptance is considered.
My comments are listed below:
In this paper, Freeland et al., conducted a qualitative study based on seventeen in-depth interviews with experts working for the African Immigrant communities and suggested major and minor barriers that restrict HBV screening and prevention for immigrant populations. Crossing the barriers will figure out the authentic number of immigrants infected with chronic HepB and offer a guidance when and how to treat existing HBV immigrates, which contributes to the aim, the elimination of Hep B by 2030 called by the WHO. In general, the paper is well organized and written, however several points require clarification before an acceptance is considered.
Major comments:
1. Line 64, 75 and 96-100, I wonder how you selected the experts across the states (N=17). What was the criteria? How about the locations and are they representative for the whole states? Western? Eastern? Is it confidential so that you did not depict more details? Also were all the participants/experts (N=17) agreed that you publish their private statements in telephone conferences?
2. Line 104-108, as a possible theme whether income scale of the African Immigrates (AI) and the availability of their social insurance also affect their negative attitude towards HBV prevention and screening?
Minor points:
Line 35, it is better to specify “The highest rates (> 5.0 %) according to the reference No. 1.
Line 37, the author cited an old publication (Kowdley et al.) and estimated 54,000 people with HBV immigrate to the states. To my knowledge, there is a recent paper (Le MH et al. Hepatology, 2019). Please cite this one as well and re-calculate the number above accordingly.
Line 75, change Appendix to Supplementary materials.
In figure 1, it would be better to implement the N value in each circle. For instance, Religious Values N=17, etc..
Line 162, I argue whether language barrier is a real impact factor affecting the AI’s communication, given that Hispanic- and Asian-Americans speak Spanish and Cantonese, respectively. Re-evaluate please the impact of language usage.
Line 195, change high blood pressure to hypertension.
Line 287, reference here No. 13 or actually 14?
Line 326, add: PLoS One 2012, 7(9): e44611
Line 335-336, I cannot find the reference No. 16. Mark it somewhere in the full-text.
Sincerely,
The reviewer
Author Response
Response to reviewer 2: Thank you for carefully reviewing our manuscript. We agree with much of the comments and suggestions that were made by reviewer 2 and have adjusted the manuscript to reflect these suggestions. Below are responses for each comment made by the reviewer 2.
Reviewer 2: Major comments:
- Line 64, 75 and 96-100, I wonder how you selected the experts across the states (N=17). What was the criteria? How about the locations and are they representative for the whole states? Western? Eastern? Is it confidential so that you did not depict more details? Also were all the participants/experts (N=17) agreed that you publish their private statements in telephone conferences?
Response to reviewer 2: Thank you for your comment. We have addressed this issue by adding clarification on the inclusion criteria in the recruitment section as well as adding specific details of participants while not compromising their identity. To clarify and provide more detail on the recruitment and the sample itself, we added more detail. Participants were selected through convenience sample from the CHIPO coalition. Inclusion criteria for participants was also updated to reflect more detail of participant selection in lines 67-75.
- Line 104-108, as a possible theme whether income scale of the African Immigrates (AI) and the availability of their social insurance also affect their negative attitude towards HBV prevention and screening?
Response to reviewer 2: Thank you for your comments. The topics of income and insurance status were not specifically brought up by the interviewees in a robust way. The themes for our analysis came directly from the interview data, and there was not enough data on insurance or income to support these as themes. While we agree that income and insurance likely play a role, the data from this study did not identify these as challenges.
Minor points:
- Line 35, it is better to specify “The highest rates (> 5.0 %) according to the reference No. 1.
Response to reviewer 2: Thank you for this comment, line 35 has been updated to reflect this change.
- Line 37, the author cited an old publication (Kowdley et al.) and estimated 54,000 people with HBV immigrate to the states. To my knowledge, there is a recent paper (Le MH et al. Hepatology, 2019). Please cite this one as well and re-calculate the number above accordingly.
Response to reviewer 2: Thank you for this comment and updated citation has been modified within the citations.
- Line 75, change Appendix to Supplementary materials.
Response to reviewer 2: Thank you for this comment, the suggested change has been made.
- In figure 1, it would be better to implement the N value in each circle. For instance, Religious Values N=17, etc..
Response to reviewer 2: Thank you for this comment. We have added a clarifying point noting that each theme displayed in figure 1 came across in all interviews.
- Line 162, I argue whether language barrier is a real impact factor affecting the AI’s communication, given that Hispanic- and Asian-Americans speak Spanish and Cantonese, respectively. Re-evaluate please the impact of language usage.
Response to reviewer 2: Thank you for your comment. Multiple studies have documented that language (non-English proficiency) serves as a major barrier for accessing care and health literacy, for most foreign-born populations. Many different languages are spoken across the African continent, and English may not be a first or second language for some AIs, inevitably affecting the ability to communicate and understand health concerns and needs. Based on interviews, this was a finding that was mentioned as a major barrier by those interviewed. Removing it would change the results of the study, and would not accurately represent the data collected from the interviews. Since the study is qualitative and based on interviews, the authors feel it is important to include.
- Line 195, change high blood pressure to hypertension.
Response to reviewer 2: Thank you for your comment, this change has been made.
- Line 287, reference here No. 13 or actually 14?
Response to reviewer 2: Thank you for your comment, this reference has been updated.
- Line 326, add: PLoS One 2012, 7(9): e44611
Response to reviewer 2: This addition has been made.
- Line 335-336, I cannot find the reference No. 16. Mark it somewhere in the full-text.
Response to Reviewer 2: This has been updated and adjusted to reflect the citation. Thank you for this comment.
Reviewer 3 Report
Catherine Freeland and colleagues conducted a qualitative study, interviewing 17 participant to investigate barriers to HBV care in African communities in the United States. The topic is very important. I have several concerns about the methodology and data analysis with potential impact on validity of the findings. More detailed comments come as follow:
Major concerns:
- Methods: Although the authors mentioned that they recruited the interviewees from CHIPO, the process of recruitment is not clear. For example, more details are required about who exactly from CHIPO the authors approached (target population); how they approached their potential participants; how many people they approached, etc.
- Methods (page 2): “Four investigators conducted seventeen individual in-depth interviews with community health experts working in AI communities across the U.S.”. This sentence implied that all interviewees were “health experts”, however, it was mentioned later that some interviewees were “community members”, and not necessarily health experts.
- Results: More information on characteristics of the participants would be informative. For example, I am willing to know how many of participants were from an African background/ethnicity and how many of participants spoke at least one African language.
- Results, Overarching themes (pages 3,4): Both overarching themes (i.e., religious values, and cultural norms) have been described in a way, implying that the authors may have assumed similar religion and culture across African communities. Religions and cultures are highly variable across African countries and even within-country in some African countries which should be addressed in data analysis.
- Results, Perception towards prevention (page 5): It was surprising that no data on HBV vaccination, as the main part of HBV prevention, have been presented in this section.
Minor concerns:
- “Rate” has been used mistakenly in place of “prevalence” in several parts of the paper. For example: “The highest rates of HBV are found in Africa and the Western Pacific regions” (page 1); “ the highest rates were in Sudan (18.6%)” (page 2).
- Introduction (page 2): “Studies have found low HBV screening rates within AI communities”. This statement should be backed up by a valid reference.
- Method (page 2): “In order to meet … programming or community outreach among AIs”. Please clarify “programming or community outreach”.
- Methods (page 2): “A literature review was conducted to identify constructs of importance for the interviews”. This sentence needs more clarification.
- Please specify what the interviews language was.
Author Response
Reviewer 3 Major concerns:
- Methods: Although the authors mentioned that they recruited the interviewees from CHIPO, the process of recruitment is not clear. For example, more details are required about who exactly from CHIPO the authors approached (target population); how they approached their potential participants; how many people they approached, etc.
Response: Thank you for your comment. We have updated the recruitment section to include more clarifying details on recruitment and the participants in this study.
- Methods (page 2): “Four investigators conducted seventeen individual in-depth interviews with community health experts working in AI communities across the U.S.”. This sentence implied that all interviewees were “health experts”, however, it was mentioned later that some interviewees were “community members”, and not necessarily health experts.
Response: Thank you for this comment. We acknowledge that this wording was confusing and we have made an adjustment to clarify this point. All members were expert physicians, public health professionals and researchers and also identified as being of African-descent. This was adjusted within the text under participant section line 133.
- Results: More information on characteristics of the participants would be informative. For example, I am willing to know how many of participants were from an African background/ethnicity and how many of participants spoke at least one African language.
Response: Thank you for this comment. We have added more detailed information on the countries served from those interviewed and more details about the participants themselves that we believe addresses this comment.
- Results, Overarching themes (pages 3,4): Both overarching themes (i.e., religious values, and cultural norms) have been described in a way, implying that the authors may have assumed similar religion and culture across African communities. Religions and cultures are highly variable across African countries and even within-country in some African countries which should be addressed in data analysis.
Response: Thank you for this important point. We certainly want to acknowledge the diversity within the African context, and we have added information in the results section lines 147-152 to highlight this.
- Results, Perception towards prevention (page 5): It was surprising that no data on HBV vaccination, as the main part of HBV prevention, have been presented in this section.
Response: Thank you for this comment. We have updated the discussion section to include some minor details on the vaccination rates for the African community.
Minor concerns:
- “Rate” has been used mistakenly in place of “prevalence” in several parts of the paper. For example: “The highest rates of HBV are found in Africa and the Western Pacific regions” (page 1); “ the highest rates were in Sudan (18.6%)” (page 2).
Response: Thank you for this comment. This has been adjusted within the introduction.
- Introduction (page 2): “Studies have found low HBV screening rates within AI communities”. This statement should be backed up by a valid reference.
Response: Thank you for this comment. This sentence has been adjusted to include citations for this statement.
- Method (page 2): “In order to meet … programming or community outreach among AIs”. Please clarify “programming or community outreach”.
Response: Thank you for this comment, language has been added to clarify programming and community outreach specifically for AI.
- Methods (page 2): “A literature review was conducted to identify constructs of importance for the interviews”. This sentence needs more clarification.
Response: This sentence has been adjusted for clarification.
- Please specify what the interviews language was.
Response: Thank you for this comment. Language was English for all interviews, this has been added within the participants section.
Round 2
Reviewer 1 Report
The authors mainly specify the usefulness of qualitative studies and no significant improvements have been added to the paper. On my opinion, the paper remains not suitable for publication in Viruses.